# Two Non-Necrotic Disease Resistance Types Distinctly Affect the Expression of Key Pathogenic Determinants of *Xanthomonas euvesicatoria* in Pepper

**DOI:** 10.3390/plants12010089

**Published:** 2022-12-24

**Authors:** Zoltán Bozsó, Dániel Krüzselyi, Ágnes Szatmári, Gábor Csilléry, János Szarka, Péter G. Ott

**Affiliations:** 1Plant Protection Institute, ELKH Centre for Agricultural Research, Herman Ottó Str. 15, H-1022 Budapest, Hungary; 2Institute of Organic Chemistry, ELKH Research Centre for Natural Sciences, Magyar Tudósok Krt 2, H-1117 Budapest, Hungary; 3Budakert Ltd., 1114 Budapest, Hungary; 4Pirospaprika Ltd., 1222 Budapest, Hungary

**Keywords:** *Xanthomonas euvesicatoria*, *Capsicum annum*, general defense system, *bs5*, *dpsA*, *hrp/hrc* genes, symptomless plant defense, pattern-triggered immunity

## Abstract

Pepper (*Capsicum annuum* L.) carrying the *gds* (corresponding to *bs5*) gene can prevent the development of bacterial leaf spot disease without HR. However, little is known regarding the development of the resistance mechanism encoded by *gds*, especially its influence on the bacterium. Here, the effect of *gds* was compared with pattern-triggered immunity (PTI), another form of asymptomatic resistance, to reveal the interactions and differences between these two defense mechanisms. The level of resistance was examined by its effect on the bacterial growth and *in planta* expression of the stress and pathogenicity genes of *Xanthomonas euvesicatoria*. PTI, which was activated with a *Pseudomonas syringae hrcC* mutant pretreatment, inhibited the growth of *Xanthomonas euvesicatoria* to a greater extent than *gds*, and the effect was additive when PTI was activated in *gds* plants. The stronger influence of PTI was further supported by the expression pattern of the *dpsA* bacterial stress gene, which reached its highest expression level in PTI-induced plants. PTI inhibited the *hrp/hrc* expression, but unexpectedly, in *gds* plant leaves, the *hrp/hrc* genes were generally expressed at a higher level than in the susceptible one. These results imply that different mechanisms underlie the *gds* and PTI to perform the symptomless defense reaction.

## 1. Introduction

Although resistance is the best way to protect plants from bacterial diseases, little is known regarding the effector mechanisms that are deployed against their pathogens. After infection, the bacteria have to cope with different levels of the plant’s defense systems. Some of these defense systems are asymptomatic, while others are conspicuous, e.g., showing visible cell death. Once a bacterium enters the plant tissues, the first activated defense system is pattern-triggered immunity (PTI) [1,2]. The elicitors of the PTI are conserved bacterial (nonself) molecules (microbe associated molecular patterns, MAMPs), which can be recognized by plant receptor-like proteins located in the plant cell membrane [3,4]. This explains why PTI can be activated by both pathogenic and nonpathogenic bacteria [5]. After recognition, signaling processes are activated, leading to various defense-related mechanisms in the plant cells, such as cell wall modifications, reactive oxygen accumulation, stomatal closure, production of putative antibacterials and hormones for posterior defense control [6], removal of apoplastic sugars [7], partly directed by transcriptional remodeling [8]. Host responses to *Xanthomonas* were recently reviewed [9]. PTI is usually asymptomatic or causes only mild symptoms in plant tissues and may stop the development of a second infection [5,10]. Previous studies have indicated that PTI impacts bacterial pathogenicity-associated gene activation (type III secretion system, T3SS) [11]. It was also shown that PTI could inhibit T3SS effector protein translocation into plant cells via an unknown mechanism [10,12].

In contrast, the other archetype of plant defense reaction, ETI (effector-induced immunity), usually involves rapid plant cell death (hypersensitivity reaction, HR). It can only be induced by live and active pathogenic bacteria [13]. The ETI/HR induction is activated inside plant cells by the bacterial protein effector(s) translocated through the T3SS and recognized by intracellular plant R resistance genes [14]. Controlled and contained plant cellular death, as a process itself, likely widens the opportunity to derange the orchestration of the virulence mechanisms of biotrophic and hemibiotrophic pathogens. Lastly, ETI depends on PTI constituents, and PTI is at a higher level under ETI [15].

One of the most important diseases of pepper is the bacterial spot disease caused by different *Xanthomonas* spp. In the absence of effective agrochemicals and cultural practices, the cultivation of resistant plants is the only solution to control this disease. During breeding programs, different resistance genes have been introduced into pepper plants. Some of them (e.g., *Bs1*, *Bs2*, *Bs3*, and *Bs4*) are based on the HR-type resistance and are determined by single dominant genes [16]. For HR-type resistance, the *Xanthomonas* effector gene and the respective plant R gene have already been identified in several cases. For example, *Bs2* encodes a nucleotide-binding site-leucine-rich (NBS-LRR) class of resistance gene [17]. The matching bacterial effector AvrBs2 has a glycerol-phosphodiesterase domain necessary for virulence function [18]. Unlike the HR-type resistances, *bs5* and *bs6* provide protection without cell death and are controlled by single recessive genes. Since the plant responses determined by the bs5 and bs6 genes not only inhibit the appearance of disease symptoms but also the multiplication of *Xanthomonas*, they can therefore be considered as resistance mechanisms [19,20]. Previously, Szarka and Csilléry also described a form of non-HR-type disease resistance in pepper to *Xanthomonas*, attributable to a monogenic recessive gene, which they named *gds* [21]. The source of resistance (PI 163192 pepper line) is the same for *gds* as for *bs5* [22]. There is only limited information about *bs5*. A patent [23] specifies the *bs5* (under the name *xcv-1* [24]) as a tail-anchored (TA) transmembrane (TM) protein. They found that the resistant gene variant carries a double leucine deletion. The bacterial elicitor of the *bs5*-governed resistance has not yet been published. Since the source of resistance and the phenotype are identical, it is very likely that *bs5* and *gds* genes are identical. This assumption is supported by crossing experiments between *gds* and *bs5* plants [25].

Strict regulation of the expression of bacterial genes directly or indirectly involved in virulence in plant tissues is required for successful infection. Hrp (hypersensitivity and pathogenicity) genes play a crucial role in successful colonization and plant defense reaction induction. They encode the proteins of T3SS, some regulators, and effectors [26]. In *Xanthomonas*, the key regulators of the *hrp* system are HrpG and HrpX. The *hrpG* gene product is an OmpR family transcriptional regulator (part of a two-component system) that directly binds to the promoter region of *hrpX*. The HrpG/X system senses the plant environment, and HrpX switches on the expression of other *hrp* genes [27]. A further important regulator of the *Xanthomonas* virulence is the RpfC/RpfG quorum-sensing system. After the cell density-dependent recognition of the cis-unsaturated fatty acid DSF by RfpC, the RpfG degrades the cyclic di-GMP signal molecule [28]. This system controls, for example, the production of exopolysaccharide (EPS), extracellular enzymes and siderophore production, biofilm formation, and motility [29]. It was also observed that there are some interactions between the HrpG/X and RpfC/G systems to fine-tune bacterial virulence [26,30,31].

The usefulness of a disease resistance correlates with its ability to inhibit the pathogen in some way. Therefore, we reasoned that plant resistance responses, such as those of PTI, ETI, or *gds/bs5*, might imprint a characteristic signature of bacterial gene expression. To explore this notion, several bacterial genes of *Xanthomonas euvesicatoria* involved in pathogenicity and virulence, such as (1) T3SS control genes, (2) T3SS structure genes, and (3) stress response genes, were chosen. We recorded the bacterial growth and expression of these selected genes *in planta* to compare the effects of said resistance mechanisms.

## 2. Materials and Methods

### 2.1. Bacteria and Bacterial Treatments of Leaves

The *Xanthomonas euvesicatoria* SZJ01 strain (referred to as *X. euvesicatoria* in this article), isolated from pepper leaves in Hungary (by János Szarka), was used to infect the plant leaves. The identity of this *Xanthomonas* was proved by PCR (Appendix A) according to [32]. *Xanthomonas euvesicatoria* KFB1 strain were used as a reference strain in PCR reactions [33]. *X. euvesicatoria* was cultured at 28 °C on YDC-agar (yeast extract, 10 g; dextrose, 20 g; CaCO_3_, 20 g; agar, 20 g; in 1000 mL) for 64 h before inoculation. For the inoculation, *Xanthomonas* suspensions in sterile tap water were adjusted to 10^8^ cells mL^−1^. 

To activate PTI, 24 h before the *X. euvesicatoria* inoculation, heat-killed *Pseudomonas syringae* pv. *syringae*
*hrcC* HR-negative mutant (*P. syringae 61 hrcC* 61-1530B, mutated by Tn*phoA*, provided by Alan Collmer, Cornell University, Ithaca, NY, USA) was injected into the leaves. *P. syringae 61 hrcC* bacteria were incubated overnight at 28 °C on King’s medium B [34]. *P. syringae 61 hrcC* were suspended in sterile water and adjusted to 10^8^ cells mL^−1^. To obtain heat-inactivated *P. syringae 61 hrcC,* the bacterial suspension was treated at 70 °C for 12 min and finally cooled down to room temperature before inoculation. Hypodermic syringes fitted with a 26 gauge needle were used to infiltrate the abaxial side of the three youngest fully developed leaves of a plant with bacterial suspensions. An approximately 30–35 cm^2^ area was infiltrated per leaf. The border of the inoculation areas was marked by a nontoxic ink immediately after the inoculations.

### 2.2. Plants Materials

A double haploid, near-isogenic bell pepper (*Capsicum annum* L.) was used for our experiments. The source of the *gds* was the *C. annuum* PI 163192 line [22,35,36]. Resistant lines were produced from the two lines by several back-crossings. Following the last back-crossings, isogenic dihaploid (DH) lines were produced based on Dumas De Vaulx et al.’s (1981) method [37]. The DH line (DH-99-71) susceptible to *X. euvesicatoria* bacterium (XS) and the DH line (DH-99-269) containing *gds* resistance gene (GDS) were used. The plants were raised in a glasshouse for 2–2.5 months in soil (general potting mix from peat, clay, and cow manure) until they reached the desired phenophase (i.e., flowering). One to two days before the first inoculations, the plants were put under controlled conditions into a growth chamber (25 °C, 80% relative humidity, 16 h daily illumination). The plants were maintained under these conditions in the growth chamber during the experiments.

### 2.3. In Planta Bacterial Growth Determination

For the inoculation, *X. euvesicatoria* suspensions were diluted from 10^8^ cells mL^−1^ with sterile tap water to 10^6^ cells mL^−1^. At 0, 6, and 24 h post-inoculation (hpi), 6 pieces of 10 mm diameter discs from three adjacent leaves of one plant were taken and pooled. At one time point, the samples were collected from three different plants to obtain three repetitions. The leaf discs were ground in a 10 mM potassium phosphate buffer at pH 7.0 with a mortar and pestle and using the plate-count technique with peptone agar plates (peptone from meat, 11 g; NaCl, 5 g; agar, 17 g; in 1000 mL), the number of viable cells per leaf area was calculated. Three biological replicates were performed using three different plant generations. The statistical analyses were carried out using the Statistica 13 software (TIBCO Software, Palo Alto, CA, USA). Analysis of variance (ANOVA) and Tukey’s post hoc test were employed, and differences at *p* < 0.05 were considered statistically significant.

### 2.4. RNA Processing and Quantitative PCR Assays

The samples for the RNA processing were taken in the same manner (size and number per treatments) for the bacterial growth determination. The leaf discs were ground in liquid nitrogen, and the total RNA was isolated with the RNeasy^®^ Protect Bacteria Mini Kit (QIAGEN, Hilden, Germany). DNase-treated (TURBO DNA-free Kit, Thermofisher, Waltham, MA, USA) total RNA (1.5 µg) was used for the synthesis of 20 μL cDNA (High-Capacity cDNA Reverse Transcription Kit, Applied Biosystems, Waltham, MA, USA) with random primers. A total of 2.5 µL from a 10-fold dilution of cDNA stock was used in each 15 µL reaction using the SensiFAST SYBR No-ROX Kit (Bioline, Memphis, TN, USA) real-time PCR mix. The final primer concentrations in a 15 µL PCR reaction were 0.3 µM. The real-time PCR amplifications were performed in a DNA Engine Opticon 2 thermocycler (MJ Research, Reno,, NV, USA). The cycling parameters were 95 °C for 3 min, followed by 40 cycles of 95 °C for 10 sec and 60 °C for 30 s. To verify the presence of single PCR products, agarose gel electrophoresis was carried out, and melting curve runs were also performed at the end of each PCR reaction. To provide negative controls, a subset of PCR reactions was carried out with noninoculated gds and XS (susceptible) samples. The PCR reactions of these samples did not provide any specific PCR products. Three technical replicates were used for each sample, and the three Ct values were averaged (i.e., arithmetic mean). The measured target gene Ct values were normalized to the *atpD* and *rpoB* reference gene Ct values [38], and the two relative gene expression values were averaged. The target sequences for the primer design were taken from the *X. euvesicatoria* strain LMG930 (GenBank: CP018467.1). Table 1 contains the sequences of the used primers. Two different plant generations were sampled for the RT-PCR analysis. The statistical analyses were carried out using the Statistica 13 software (TIBCO Software, Palo Alto, CA, USA). Analysis of variance (ANOVA) and Tukey’s post-hoc test were employed, and differences at *p* < 0.05 were considered statistically significant.

## 3. Results and Discussion

### 3.1. X. euvesicatoria Multiplies to Varying Degrees in Plants with Different Resistance Responses

To follow the bacterial cell number changes in different *X. euvesicatoria*–pepper interactions, the bacteria were injected at 10^6^ cells mL^−1^ into susceptible (XS) and resistant (GDS) pepper leaves. PTI was activated in leaves by pre-injection with heat-killed *P. syringae* 61 *hrcC* 24 h prior to the *X. euvesicatoria* inoculation. The number of *X. euvesicatoria* cells in the leaves was determined after 0, 6, and 24 h post-inoculations (hpi). Between 0 and 6 hpi, a decrease in the number of *X. euvesicatoria* cells was observable, even in the susceptible host, suggesting that a temporary period in the apoplast of leaves is necessary for the bacteria to adapt to the plant environment. By 24 hpi, the *X. euvesicatoria* cells multiplied at different rates and reached different cell numbers depending on the activated resistance. The highest *X. euvesicatoria* cell numbers were found in the susceptible XS plants, and lower cell numbers could be re-isolated from the GDS plants. A previous activation of PTI caused a decrease in the culturable cell number both in XS and in GDS leaves at 24 hpi. The lowest cell counts came from the GDS plus PTI treatments, suggesting that *gds* and PTI have a synergistic effect (Figure 1). There were no visible symptoms on the plant leaves during the experiments (0–24 hpi), although the infections of the gds plants with *X. euvesicatoria* [16] or the induction of PTI by heat-killed *P. syringae 61 hrcC* could cause late slight yellowing of the inoculated area after one or two weeks. Pathogenic bacterial survival within a resistant plant is a summation, reflecting multiple host factors acting in a complex way. The most straightforward interpretation would be that *gds* and PTI use partially different mechanisms to inhibit bacteria. While in the PTI treatment the effects of an already well-developed PTI is to be expected, the impact on bacteria by GDS plants is largely unknown, and in all three treatments the PTI must also be induced by MAMPs of the pathogen. The synergism between PTI and GDS either points to an enhanced PTI (e.g., as it was shown during ETI, see [15]) or that *gds* and PTI use partially different mechanisms to inhibit bacteria.

### 3.2. In Planta Expression of Stress-Related dpsA Gene

Dps is a nonspecific DNA-binding protein that involves in the protection of cells from various stresses, such as oxidative, thermal, acid, and base stress; UV and gamma irradiation; and iron and copper toxicity [34]. To determine whether different resistance responses induce different levels of stress responses in *X. euvesicatoria*, we measured the *in planta* expression of *dpsA* (Figure 2). While at all time points the *dpsA* transcription was low in the susceptible host, in agreement with the supposition that this is a relatively optimal environment for a virulent pathogen, the *dspA* expression increased to varying degrees in the PTI-activated and GDS plants. PTI increased *dpsA* expression at 6 hpi both in the XS and GDS plants, but later, this effect strongly decreased (GDS + PTI) or disappeared (XS + PTI). On the other hand, in GDS plant leaves the amount of *dpsA* transcripts increased later, at 24 hpi. In accordance, the aggregate effect of *gds* and PTI seemed interposed at 24 hpi (Figure 2). Based on these results, the stress caused by PTI seems associated with the early and that by *gds* with the late phase.

### 3.3. In Planta Expression of T3SS-Related Regulators

The T3SS system is controlled by internal (e.g., quorum signal and cyclic di-GMP) and external (e.g., host) cues. The RpfC and RpfG regulators are more directly related to internal signals, while the HrpG-HrpX signaling axis responds more to the host environment [27,28]. We found that *hrpG* and *hrpX* transcription responded to our treatments more characteristically than *rpfC* and *rpfG* (Figure 3). Both *hrpG* and *hrpX* had a low basal expression level at 0 hpi and were induced within the plant by 6 hpi. However, compared to the susceptible host, the *gds* trait stimulated *hrpG* transcription at 6 hpi and even more at 24 hpi. The activation of PTI was able to suppress it at 24 hpi both in susceptible (XS) and resistant (GDS) leaves. PTI was also inhibitory to *hrpX* transcription, while *gds* had no influence since there was no significant enhancement compared to the susceptible combination. In contrast to *hrpG* and *hrpX,* no obvious pattern of gene expression changes was found for *rpfC* and *rpfG*. Interestingly, at 0 hpi, both *rfpC* and *rfpG* showed prominent expression in PTI-activated GDS leaves, which suggests a very quick, predisposed plant response that may affect bacterial gene expressions.

### 3.4. In Planta Expression of T3SS Structural Genes

The T3SS machinery is a protein complex, called Hrp pilus, consisting of a base that anchors in both the inner and outer membranes and a long hollow filament that can reach the host plasma membrane by growing through the plant cell wall. The effector proteins pass through the filament hole [26]. The expression of genes coding for structural T3SS components, such as *hcrC*, *hrcU*, *hrpE,* and *hrpF*, answered uniformly to a particular *in planta* conditions (Figure 4): As expected, their levels were low or undetectable immediately after infections (at 0 hpi) and were induced in the susceptible plant tissue that served as a reference “treatment” in our study. The data reflect the increased demand for the filament monomer (HrpE) during the early steps of the pathogenesis because *hrpE* showed the highest relative gene expression. Surprisingly, the tested structural bacterial genes were induced more in the resistant (GDS) plants than in the susceptible (XS) plant leaves, while the induction was hampered by PTI both in the XS and *gds* plants.

## 4. Conclusions

For breeders, successful plant defense is most often associated with the hypersensitive reaction (HR) involving plant cell death. In this study, we compared the effects of two symptomless, resistance responses in pepper on the important bacterial pathogen *X. euvesicatoria*. Although these defense mechanisms cannot be associated with plant cell death, they could inhibit bacterial multiplication in the intercellular spaces of leaf tissues (Figure 1). Our *in planta* bacterial gene expression results suggested that despite their phenotypic similarities, *gds* and PTI may have a different time course, eventually culminating in distinct antibacterial mechanisms. The stress marker *dpsA* gene activation was quicker in the case of PTI-pre-activated leaves than in the *gds* plants (Figure 2). The more rapid activation of defense reactions in PTI-activated plants may contribute to more effective bacterial cell number restriction (Figure 1). The PTI’s suppressing effect, both on the T3SS regulators (*hrpG/hrpX*) and structural gene expression (Figure 3 and Figure 4, respectively) could have a role here. This result is consistent with previous ones that indicated the PTI impacts on T3SS and even T3SS effector protein translocation [10,11,12]. Most intriguing is the fact that, unlike PTI, *gds* upregulated *hrp* gene transcription compared to the susceptible plant leaves (Figure 3 and Figure 4). The higher expression may be a consequence of an extra stimulus derived from the plant cells and/or the disruption of the bacterial *hrp* gene regulatory system (activation of *hrpG* regulator in *gds* plants at 24 hpi, Figure 3A), which may also lead to a decreased bacterial pathogenicity.

In summary, our results imply different antibacterial mechanisms of PTI and *gds* in pepper plants, and since they have an additive effect on bacteria, we recommend their combined use for resistance breeding.

## Figures and Tables

**Figure 1 plants-12-00089-f001:**
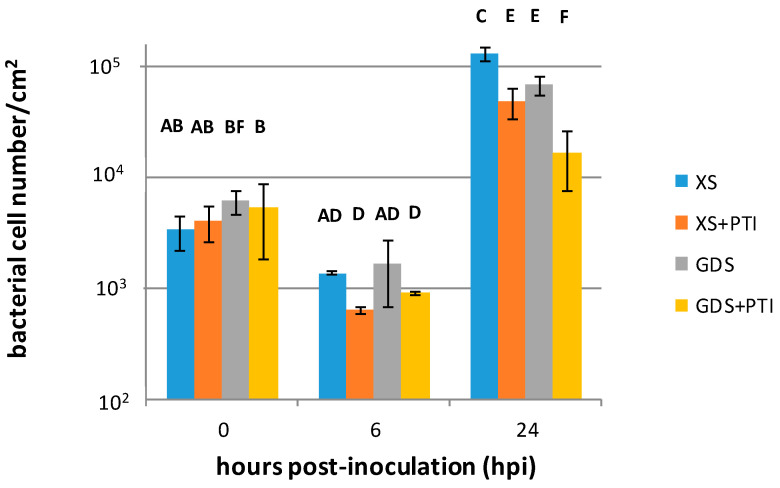
Number of *X. euvesicatoria* cells in pepper leaves as affected by resistance mechanisms. *X. euvesicatoria* was injected at 10^6^ cells mL^−1^ into pepper leaves at the 0 time point. PTI was initiated 24 h before *X. euvesicatoria* inoculation by injecting the same leaves with heat-killed *P. syringae* 61 *hrcC*. Leaf samples were taken at the indicated time points after *X. euvesicatoria* infections, and the bacterial cell number was determined by the plate-count technique. Three biological replicates were performed, and similar patterns of population dynamics were obtained. Error bars represent mean ± standard deviations of three replicates of a representative experiment. The letters above the columns indicate significantly different groups. XS: susceptible plant, GDS: *gds* gene-carrying resistant plant, +PTI: PTI activated in the plant before *X. euvesicatoria* inoculation.

**Figure 2 plants-12-00089-f002:**
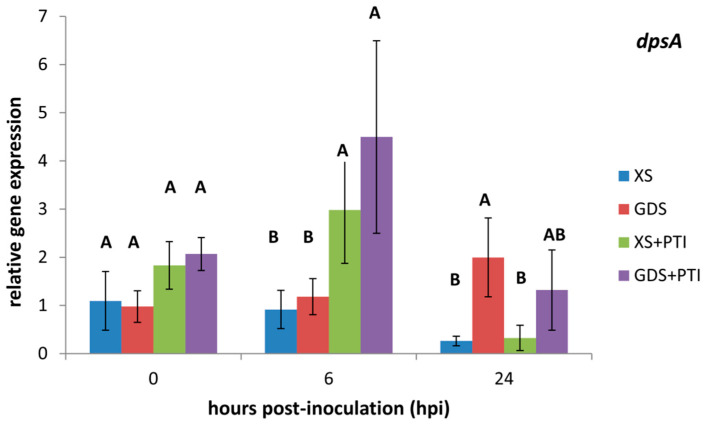
*In planta* expression of *dpsA* of *X. euvesicatoria* as affected by PTI and/or GDS. At 0 hpi, the plant leaves were injected with 10^8^ cell mL^−1^ *X. euvesicatoria*. PTI was initiated 24 h prior to *X. euvesicatoria* inoculation by injecting the same leaves with heat-killed *P. syringae* 61 *hrcC*. Leaf samples were taken at the indicated time points after *X. euvesicatoria* infections. To determine the relative gene expression, the measured target gene Ct values were normalized to those of the reference genes (*atpD* and *rpoB*), and the two relative gene expression values were averaged. Bars represent the mean ± standard deviations of two replicates. The letters above the columns indicate significantly different groups for the values belonging to the given sampling time. XS: susceptible plant, GDS: *gds* gene-carrying resistant plant, +PTI: PTI activated in the plant before *X. euvesicatoria* inoculation.

**Figure 3 plants-12-00089-f003:**
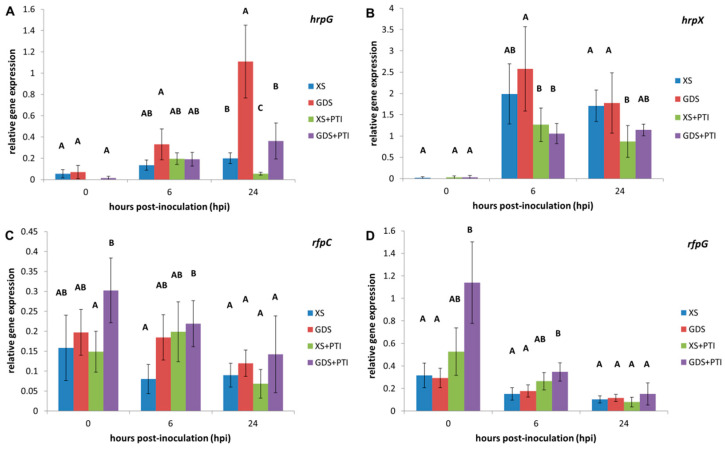
*In planta* expression of selected T3SS-related regulator genes of *X. euvesicatoria* as affected by PTI and GDS: (**A**) *hrpG;* (**B**) *hrpX;* (**C**) *rfpC;* (**D**) *rfpG*. At 0 hpi, the plant leaves were injected with 10^8^ cell mL^−1^ *X. euvesicatoria*. PTI was initiated 24 h prior to *X. euvesicatoria* inoculation by injecting the same leaves with heat-killed *P. syringae* 61 *hrcC*. Leaf samples were taken at the indicated time points after *X. euvesicatoria* infections. To determine the relative gene expression, the measured target gene Ct values were normalized to those of the reference genes (*atpD* and *rpoB*), and the two relative gene expression values were averaged. Bars represent the mean ± standard deviations of two replicates. The letters above the columns indicate significantly different groups for the values belonging to the given sampling time. XS: susceptible plant, GDS: *gds* gene-carrying resistant plant, +PTI: PTI activated in plant before *X. euvesicatoria* inoculation.

**Figure 4 plants-12-00089-f004:**
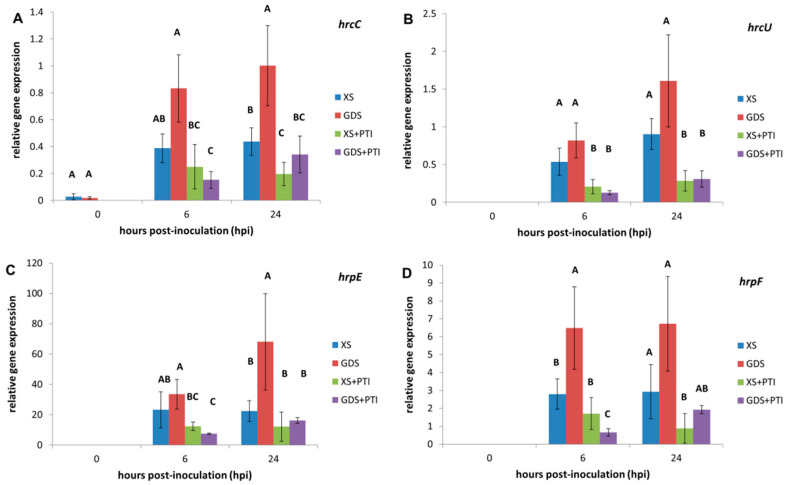
In planta expression of the T3SS structural genes of *X. euvesicatoria* as affected by PTI and GDS: (**A**) *hrcC;* (**B**) *hrcU;* (**C**) *hrpE;* (**D**) *hrpF*. At 0 hpi, the plant leaves were injected with 10^8^ cell mL^−1^ *X. euvesicatoria*. PTI was initiated 24 h prior to *X. euvesicatoria* inoculation by injecting the same leaves with heat-killed *P. syringae* 61 *hrcC*. Leaf samples were taken at the indicated time points after *X. euvesicatoria* infections. To determine the relative gene expression, the measured target gene Ct values were normalized to those of the reference genes (*atpD* and *rpoB*), and the two relative gene expression values were averaged. Bars represent the mean ± standard deviations of two replicates. The letters above the columns indicate significantly different groups for the values belonging to the given sampling time. XS: susceptible plant, GDS: *gds* gene-carrying resistant plant, +PTI: PTI activated in plant before *X. euvesicatoria* inoculation.

**Table 1 plants-12-00089-t001:** Primers used in this study to measure *Xanthomonas euvesicatoria* gene expressions.

GenBank ID ^a^	Gene	Forward Primer	Reverse Primer
BJD11_20695	*hrcC*	5’ TCAAAGAAGTGCTGCGTGAT 3’	5’ CCAGACAAAGCCGTAGGTG 3’
BJD11_20740	*hrcU*	5’ CCGATTTCATCACCAATAAC 3’	5’ AGAAGCCGACCGAGAAGAAACT 3’
BJD11_20785	*hrpE*	5’ CCGATGAACTTGTTGAGTGC 3’	5’ GACGAGGCTCAGAAGTCCAT 3’
BJD11_20815	*hrpF*	5’ GCCGATCCAGAACCGAAACA 3’	5’ AACTGGGCGGGAAGAACGAC 3’
BJD11_16105	*hrpX*	5’ GACTGCAACATCTCCAACAG 3’	5’ CTGATATTCCAGGATCAGCAAC 3’
BJD11_16110	*hrpG*	5’ CGAAGATCAGCAGCTCGCA 3’	5’ GATCGGTGTTCCTGTTGACG 3’
BJD11_12825	*rpfC*	5’ CGATCCTGATTTCGCCTTACT 3’	5’ ATCAAGCCCAGCAACAATCC 3’
BJD11_12835	*rpfG*	5’ TCGACTTCCTGGTCAAGCCGATCC 3’	5’ GCGCTCTTCGACCTCGTTCATGC 3’
BJD11_06505	*dpsA*	5’ CGTTGACCTCGATCCCGGAAGA 3’	5’ CTGACGCACCATTTCACGCCAGTC 3’
BJD11_17780	*rpoB*	5’ GGAACTGATCAATGCCAAGCC 3’	5’ TCTGGTCCATGAACTGCGAC 3’
BJD11_03830	*atpD*	5’ TACACCATCGCCACCTTGTC 3’	5’ GGCAACGACTTCTACCACGAGA 3’

^a^ *Xanthomonas euvesicatoria* strain LMG930 (GenBank:CP018467.1).

## Data Availability

All the available data are presented in the manuscript.

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
