# Peer review of "Two Non-Necrotic Disease Resistance Types Distinctly Affect the Expression of Key Pathogenic Determinants of Xanthomonas euvesicatoria in Pepper"

_plants, 2022, doi:10.3390/plants12010089_

Round 1
Reviewer 1 Report
The study presents a characterization through gene expression analysis of the differences and overlap between two different plant defense mechanisms in bell pepper: a resistance gene and PTI induction.
The results are very straightforward and clear to understand and, even though the fine molecular mechanisms behind the phenotypic response are not cleared or investigated in this study, these results do support the hypothesis of the work and are therefore considered adequate.
A major concern regarding the whole study is the apparent lack of a negative control, consisting of the two different plant backgrounds used in the study without X. euvesicatoria inoculum. Without these, it is unknown if the dpsA levels are in line with those of a healthy/uninfected plant at any point of the experiment. Likewise, it is unknown if the levels of TSIII related genes are solely due to the inoculated X. euvesicatoria strain or if bacteria native to the plant microbiome contribute to these values and possibly interfere with the accurate quantification.
The other major concern is that the plants were not analyzed for development of symptoms. This is an important evaluation that would put the results in a clearer applicative perspective… the final conclusion is that these two mechanisms have synergic effects, but this is demonstrated only by gene expression, there seems to be no demonstration that there is a sharper reduction of symptoms, or a total absence of symptoms, when using both, so the benefit of this combination is very likely effective, but it remains only a speculation without a symptom evaluation that backs the claim.
There are also several other points of critique that should be addressed regarding the study and its exposition:
1 - The concept of “resistance” to pathogen is very complex and the term sees very different definitions by different researchers. By the results, that show colonization by the pathogen in the plants with gds, I can imagine what the authors mean by resistance, but many would consider that as “tolerance” instead. For this reason, before speaking of the use of a resistant plant, the authors are advised to provide the definition of resistance that they use for this phenomenon.
2 - Lines 97-98: This definition of the strain is not sufficient. Does the used strain have a strain name/ID/code? Any deposited sequence in a database related to it? Also, what does “proven by PCR” mean? Was it a species-specific PCR, or a PCR of barcoding genes followed by sequencing? Was this particular strain proven to be pathogenic before? Considering the number of questions regarding the strain, “data not shown” is not recommended.
3 - Lines 107-109: this procedure is carried out only for P. syringae? Or for both P. syringae and X. euvesicatoria? Also, it seems like the content of lines 121-122 should go right before this sentence, rather than where it was placed. How much volume, more or less, was inoculated in the leaves?
4 - Lines 117-119: The timing here is vague… one or two days before the experiment, considering the results that show that PTI effects can drop considerably in the 6h – 24h period can make a huge difference. Likewise, 2 or 2.5 months can make a lot of difference for RNA-based experiments. Lastly, what does “and during the experiment” mean? Was P. syringae inoculated also during the experiment? If so, how many times and how often?
5 - Lines 122-123: in this sentence, and other throughout materials and methods, it is unclear how were the samples constructed: were the discs analyzed singularly, or pooled? If pooled, in which way?
6 - Line 143: the kind of mean used to average the expression levels should be expressed. Was it arithmetic mean, geometric mean, harmonic mean, other?
7 - Lines 158-159 and Figure 1: The text and figure do not seem to be in accordance. The authors state that the amount of cells “stagnates”, suggesting the level remaining the same, but the graph seems to show a sharp decrease instead. Maybe using a log10 scale for the graph might make the interpretation of the results easier and clearer.
Author Response
We would like to thank the reviewers for their helpful comments and questions that helped improve the manuscript.
Answers to reviewers’ comments and suggestions
Review 1
A major concern regarding the whole study is the apparent lack of a negative control, consisting of the two different plant backgrounds used in the study without X. euvesicatoria inoculum. Without these, it is unknown if the dpsA levels are in line with those of a healthy/uninfected plant at any point of the experiment. Likewise, it is unknown if the levels of TSIII related genes are solely due to the inoculated X. euvesicatoria strain or if bacteria native to the plant microbiome contribute to these values and possibly interfere with the accurate quantification
-We do have the data in question, i.e., from those susceptible and gds plants samples that were not infiltrated with X. euvesicatoria. The PCR reactions of these samples did not provide any specific PCR products. So the measured gene expression changes were due to alterations in the gene expression of X. euvesicatoria The original manuscript did not contained this information, therefore in the new version we completed the Materia and methods section (2.4) with this information.
The other major concern is that the plants were not analyzed for development of symptoms. This is an important evaluation that would put the results in a clearer applicative perspective… the final conclusion is that these two mechanisms have synergic effects, but this is demonstrated only by gene expression, there seems to be no demonstration that there is a sharper reduction of symptoms, or a total absence of symptoms, when using both, so the benefit of this combination is very likely effective, but it remains only a speculation without a symptom evaluation that backs the claim.
-These two plant defense reactions (PTI and gds/bs5) are symptomless. The combinations of the reactions were also symptomless. Although infections of gds plants with Xanthomonas or induction of PTI by heat-killed Pseudomonas syringae could cause late slight yellowing of the inoculated area, during the period of the experiments (0-24 hpi) there were no visible symptoms on plant leaves. We added this information to the Results and discussion section (3.1).
There are also several other points of critique that should be addressed regarding the study and its exposition:
1 - The concept of “resistance” to pathogen is very complex and the term sees very different definitions by different researchers. By the results, that show colonization by the pathogen in the plants with gds, I can imagine what the authors mean by resistance, but many would consider that as “tolerance” instead. For this reason, before speaking of the use of a resistant plant, the authors are advised to provide the definition of resistance that they use for this phenomenon.
In general, the concept of “resistance” is defined as the host’s ability to limit pathogen multiplication, while the tolerance is defined as the host’s ability to reduce the negative effects of infection (Pagán and Garcia 2020). In addition to the fact that the reaction is symptomless in pepper plants carrying gds/bs5 gene, the multiplication Xanhomonas euvesicatoria were also restricted compared to susceptible plants. At the beginning of the infections (approx. 0-2 dpi) in gds/bs5 pepper leaves the Xanthomonas bacteria multiply at a lower growth rate than in susceptible plants and finally reach about a 2-3 log lower level of bacterial populations (Vallejos et al 2010 and our unpublished result). So, the gds/bs5 gene confers resistance against bacterial pathogens. We completed the introduction section with this information.
Pagán I, García-Arenal F. Tolerance of Plants to Pathogens: A Unifying View. Annu Rev Phytopathol. 2020;58:77-96. doi: 10.1146/annurev-phyto-010820-012749.
Vallejos CE, Jones V, Stall RE, Jones JB, Minsavage GV, Schultz DC, Rodrigues R, Olsen LE, Mazourek M. Characterization of two recessive genes controlling resistance to all races of bacterial spot in peppers. Theor Appl Genet. 2010 121(1):37-46. doi: 10.1007/s00122-010-1289-6.
2 - Lines 97-98: This definition of the strain is not sufficient. Does the used strain have a strain name/ID/code? Any deposited sequence in a database related to it? Also, what does “proven by PCR” mean? Was it a species-specific PCR, or a PCR of barcoding genes followed by sequencing? Was this particular strain proven to be pathogenic before? Considering the number of questions regarding the strain, “data not shown” is not recommended.
- Xanthomonas euvesicatoria, used for infections is a Hungarian isolate, was isolated from pepper by one of the authors (János Szarka). Before starting the PCR experiments, the pathogenicity was checked and compared to a Xanthomonas euvesicatoria type strain (ATCC 11633, NCPPB 2968). Since the Hungarian isolate were more virulent in our system (multiplid better in the susceptible pepper leaves) we selected this strain for further work. The identity of this strain was proved by PCR methods according to Araújo et al. 2012. The PCR reactions use AFLP-based primer pairs that can discriminate between X. euvesicatoria, X. vesicatoria, X, perforans and X. gardneri. Similar to other reference X. euvesicatoria strain(s), specific PCR products were obtained only with X. euvesicatoria primers. The picture one the agarose gel of the PCR results was attached to the manuscript as a supplementing material.
This Xanthomonas euvesicatoria strain under the name “SZJ01” is deposited in our institute and is available to others from there.
3 - Lines 107-109: this procedure is carried out only for P. syringae? Or for both P. syringae and X. euvesicatoria? Also, it seems like the content of lines 121-122 should go right before this sentence, rather than where it was placed. How much volume, more or less, was inoculated in the leaves?
-The heat treatment carried out only with P. syringae hrcC. The sentence was modified in the text to make it clear which bacteria the treatment applies to.
-the sentence (line121-122): we would leave the sentence in its current place, because it refers to In planta bacterial growth determination part of the methods.
-The exact volume of inoculated bacterial suspension is not known because we aimed to inject approximately the same size plant area. However, in general, we can calculate with about 10 μl of inoculum per cm2 of leaf surface.
4 - Lines 117-119: The timing here is vague… one or two days before the experiment, considering the results that show that PTI effects can drop considerably in the 6h – 24h period can make a huge difference. Likewise, 2 or 2.5 months can make a lot of difference for RNA-based experiments. Lastly, what does “and during the experiment” mean? Was P. syringae inoculated also during the experiment? If so, how many times and how often?
The plants grow in a glasshouse for 2-2.5 month, where the temperature and the lighting control were limited. Therefore the plants were maintained in greenhouse not for a specific period but until they reached the desired phenophase (flowering). After that, we put the plants into a growth chamber (25 °C, 80% relative humidity, 16 h daily illumination) for one-two days before first inoculation and the plants were maintained in the growth chamber until the last RNA sample collections. The one-two days conditioning period was introduced in order to bring the plants in approximately the same condition for inoculations. The heat-killed P. syringae hrcC was inoculated after the conditioning period once at the same time into gds and XS plants. During the experiment we did not experience any difference between the reactions of plants that were one or two days in growth chamber before first inoculation.
5 - Lines 122-123: in this sentence, and other throughout materials and methods, it is unclear how were the samples constructed: were the discs analyzed singularly, or pooled? If pooled, in which way?
- At all time points 6 pieces of 10 mm diameter discs from three adjacent leaves of one plant were taken and pooled. At one time point the samples collected from three different plants to obtain three repetitions. The text was completed to clarify the method.
6 - Line 143: the kind of mean used to average the expression levels should be expressed. Was it arithmetic mean, geometric mean, harmonic mean, other?
- We used arithmetic mean. The text was completed with this information.
7 - Lines 158-159 and Figure 1: The text and figure do not seem to be in accordance. The authors state that the amount of cells “stagnates”, suggesting the level remaining the same, but the graph seems to show a sharp decrease instead. Maybe using a log10 scale for the graph might make the interpretation of the results easier and clearer.
-Between the 0 and 6 hpi there was a less pronounced difference than between 6 and 24 hpi but the decrease of Xanthomonas population 6 hours after inoculation was typical in the repeated experiments. Therefore the text was corrected according to the reviewer’s suggestion. According the reviewer nr. 2’s advice the Figure 1 was replaced with a block chart and as reviewer nr. 1 suggested the scale was changed to log10.

Reviewer 2 Report
reviewer's comments
the subject of the research were plants with the gds gene, so there is no need to mention bs5 in the abstract
information about the X.euvesicatoria isolate are needed ( origin of the isolat number, deposition place)
figure 1 should be replaced with a block chart. the lines used between the measurement time points suggested that bacterial growth was linear. It is very unlikely
line 184 the authors write that At all time points, dpsA transcription was lowest it is not like that. At time point zero, expression of dpsa was not the lowest for xs. Statistically significant differences between XS and GDS were visible only after 24 hours
Author Response
We would like to thank the reviewers for their helpful comments and questions that helped improve the manuscript.
Answers to reviewers’ comments and suggestions
Review2
-the subject of the research were plants with the gds gene, so there is no need to mention bs5 in the abstract
As the bs5 is more widely accepted name of this resistance and the gds and bs5 are the same, for easier understanding we would leave bs5 in the summary as well. (the Hungarian breeders who are co-authors of this manuscript insist on the name gds because of their priority on this field)
-information about the X.euvesicatoria isolate are needed ( origin of the isolat number, deposition place)
- Xanthomonas euvesicatoria, used for infections is a Hungarian isolate, was isolated from pepper by one of the authors (János Szarka). Before starting the PCR experiments, the pathogenicity was checked and compared to a Xanthomonas euvesicatoria type strain (ATCC 11633, NCPPB 2968). Since the Hungarian isolate were more virulent in our system (multiplid better in the susceptible pepper leaves) we selected this strain for further work. The identity of this strain was proved by PCR methods according to Araújo et al. 2012. The PCR reactions use AFLP-based primer pairs that can discriminate between X. euvesicatoria, X. vesicatoria, X, perforans and X. gardneri. Similar to other reference X. euvesicatoria strain(s), specific PCR products were obtained only with X. euvesicatoria primers. The picture one of the agarose gel of the PCR results was attached to the manuscript as a supplementing material.
This Xanthomonas euvesicatoria strain under the name “SZJ01” is deposited in our institute and is available to others from there.
-figure 1 should be replaced with a block chart. the lines used between the measurement time points suggested that bacterial growth was linear. It is very unlikely
We agree with the reviewer's suggestion and according to it the Figure 1 was replaced with block chart.
-line 184 the authors write that At all time points, dpsA transcription was lowest it is not like that. At time point zero, expression of dpsa was not the lowest for xs. Statistically significant differences between XS and GDS were visible only after 24 hours
We have rewritten the text to match the results shown in the figure.
“While at all time points the dpsA transcription was low in the susceptible host, in agreement with the supposition that this is a relatively optimal environment for a virulent pathogen, the dspA expression increased to varying degrees in PTI-activated and GDS plants.”

Round 2
Reviewer 1 Report
The authors provided sufficient data, amendment to the text, and justification of choices made to adequately respond to the concerns regarding the previous version of the manuscript.
I have no further comments regarding the manuscript.
Author Response
-